# Rep3D: Re-parameterize Large 3D Kernels with Low-Rank Receptive Modeling for Medical Imaging

## Abstract

In contrast to vision transformers, which model long-range dependencies through global self-attention, large kernel convolutions provide a more efficient and scalable alternative, particularly in high-resolution 3D volumetric settings. However, naïvely increasing kernel size often leads to optimization instability and degradation in performance. Motivated by the spatial bias observed in effective receptive fields (ERFs), we hypothesize that different kernel elements converge at variable rates during training. To support this, we derive a theoretical connection between element-wise gradients and first-order optimization, showing that structurally re-parameterized convolution blocks inherently induce spatially varying learning rates. Building on this insight, we introduce Rep3D, a 3D convolutional framework that incorporates a learnable spatial prior into large kernel training. A lightweight two-stage modulation network generates a receptive-biased scaling mask, adaptively re-weighting kernel updates and enabling local-to-global convergence behavior. Rep3D adopts a plain encoder design with large depthwise convolutions, avoiding the architectural complexity of multi-branch compositions. We evaluate Rep3D on five challenging 3D segmentation benchmarks and demonstrate consistent improvements over state-of-the-art baselines, including transformer-based and fixed-prior re-parameterization methods. By unifying spatial inductive bias with optimization-aware learning, Rep3D offers an interpretable, and scalable solution for 3D medical image analysis. The source code is publicly available at ****.

## 1 Introduction

The landscape of medical vision models has evolved rapidly, expanding from early convolutional architectures to modern transformer-based designs. In particular, Vision Transformers (ViTs) have gained traction for their ability to model long-range dependencies using multi-head self-attention and minimal inductive bias Dosovitskiy et al. (2020). In parallel, the community has revisited large kernel convolutions as a scalable alternative to attention mechanisms, particularly in the context of high-resolution 3D volumetric data Liu et al. (2022b); Lee et al. (2022). Despite architectural differences, both ViTs and large-kernel CNNs share a central goal: expanding the effective receptive field (ERF) to enable rich spatial context aggregation. However, simply increasing kernel size does not guarantee improved performance. Prior work has shown that naïve enlargement of convolutional filters can result in saturated or degraded accuracy across various segmentation tasks Ding et al. (2022b); Lee et al. (2023). Unlike ViTs, which adaptively attend to spatial content, standard convolutions rely on static, weight-shared kernels and lack the ability to modulate importance across spatial positions. This limitation prompts our first research question: **Can we incorporate spatial priors into large kernel convolutions to improve learning effectiveness?**

Recent advances in structural re-parameterization offer a promising direction. Methods such as RepLKNet Ding et al. (2022b), SLaK Liu et al. (2022a), and PELK Chen et al. (2024) scale kernels to extreme sizes (e.g., $31 \times 31$, $51 \times 51$, $101 \times 101$) by combining parallel branches of "large + small" convolutions into what is referred to as a Constant-Scale Linear Addition (CSLA) block. These parallel paths are merged into a single kernel at inference time, enabling efficient deployment while capturing multi-scale features during training. Interestingly, we observe that CSLA blocks naturally encode spatial learning bias: elements near the kernel center tend to converge faster than those on

Figure 1: (a) Traditional structural re-parameterization methods (e.g., CSLA blocks) re-parameterize small and large kernel convolutions to improve representational capacity, but apply linear optimization with same learning rate across the kernels, demonstrating a faster convergence in local then global. (b) In contrast, Rep3D introduces a learnable spatial bias via a generator network $f_\theta$, which modulates each element in the large kernel using a prior based on distance decay. This adaptive modulation enables local-to-global update dynamics aligned with ERF behavior, enhancing both training stability and model performance for 3D volumetric tasks.

the periphery. This mirrors diffusion-like gradient propagation in ERFs starting from the center and expanding outward. These observations suggest that convergence dynamics are not uniform across the kernel, but instead spatially structured. This leads to our second question: **Can we explicitly model this diffusion pattern as a learnable spatial prior to re-weight kernel element updates during training?**

To address this, we first provide a theoretical analysis of the optimization dynamics in CSLA-based re-parameterized convolutions. We show that each branch (e.g., small vs. large kernels) can implicitly operates under a distinct learning rate, leading to element-wise differences in convergence speed. These dynamics correlate with ERF visualizations and share characteristics with spatial frequency patterns in human visual perception Kulikowski et al. (1982). Inspired by this, we propose a novel receptive bias re-parameterization strategy that encodes spatial distance from the kernel center as a spatial bias prior on learning convergence. We implement this as a low-rank modulation mechanism that generates spatial scaling factors for kernel weights, allowing the optimizer to emphasize local versus global regions adaptively for gradient back-propagation. Building on this insight, we present Rep3D, a 3D convolutional architecture that integrates large kernel convolutions (e.g., $21 \times 21 \times 21$) with our proposed re-parameterization approach. Unlike prior approaches that rely on multi-branch structures, Rep3D employs a plain and efficient encoder to reduce complexity while preserving representational capacity. We evaluate Rep3D across five challenging volumetric medical segmentation benchmarks and show that it consistently outperforms state-of-the-art transformer- and CNN-based models.Our key contributions are as follows:

- We propose Rep3D, a 3D CNN with large kernel convolutions and a streamlined encoder design that achieves state-of-the-art (SOTA) performance on multi-scale (i.e. from organs/tissues to tumors) segmentation benchmarks.

- We propose a novel and theoretically grounded re-parameterization approach that models ERF diffusion as a learnable spatial bias prior, enabling element-wise modulation of gradient convergence for training.

- We validate our method on five challenging 3D medical imaging benchmarks under direct training settings, achieving consistent and significant improvements across all datasets.

## 2 RELATED WORK

**CNN-based 3D Models:** Foundational architectures such as 3D U-Net Çiçek et al. (2016), V-Net Milletari et al. (2016), and nnU-Net Isensee et al. (2021) have played a pivotal role in establishing the standard for volumetric medical image segmentation. These models rely on encoder-decoder designs with dense skip connections, offering a strong balance between spatial resolution and semantic representation. Due to their stability, interpretability, and effectiveness without requiring large-scale pre-training, they remain widely used in clinical and research benchmarks. However, their inherently local receptive fields limit their ability to capture long-range dependencies, motivating subsequent architectural innovations aimed at expanding the effective receptive field (ERF).

**Vision Transformer-based and Hybrid Models:** To overcome the locality constraints of CNNs, transformer-based architectures such as UNETR Hatamizadeh et al. (2022b) and SwinUNETR Hatamizadeh et al. (2022a) introduce global self-attention mechanisms that model distant spatial dependencies more effectively (i.e. Follow-up models like nnFormer Zhou et al. (2021), Swin-Unet Cao et al. (2021), and SwinBTS Jiang et al. (2022)). These models encode context across entire volumes through hierarchical token representations, marking a major shift in design philosophy. However, they typically require large-scale pre-training and introduce significant computational complexity due to the quadratic scaling of attention, particularly problematic in 3D volumetric settings. Additionally, their reliance on patch-based tokenization can compromise fine-grained spatial precision—crucial in dense prediction tasks like medical segmentation.

**Large Kernel Convolution Networks:** A more recent and efficient alternative to transformers involves expanding the ERF through large kernel convolutions, as demonstrated by models such as ConvNeXt Liu et al. (2022b), 3D UX-Net Lee et al. (2022), and MedNeXt Roy et al. (2023). These architectures leverage depth-wise or separable convolutions to approximate global context modeling while preserving the simplicity and inductive biases of convolutional designs. However, studies in 2D vision backbone(e.g., RepLKNet Ding et al. (2022b) (kernel size: $31 \times 31$), SLaK Liu et al. (2022a) (kernel size: $51 \times 51$)) reveal that naively scaling up kernel size leads to saturation or performance degradation in the absence of additional structural guidance. This key insight motivates the design of Rep3D, which augments large 3D kernels with a learnable spatial prior inspired by ERF theory. By explicitly guiding convergence dynamics across kernel elements, Rep3D enables more effective utilization of large kernels, bridging the gap between CNN efficiency and transformer-like contextual modeling.

**The integration of Weight Re-parameterization.** Structural re-parameterization (SR) has emerged as a powerful paradigm to enhance CNN training without altering inference-time complexity. Models like RepVGG Ding et al. (2021) and OREPA Hu et al. (2022) employ additional convolution branches (e.g., $1 \times 1$ or identity paths) during training to improve gradient flow and feature diversity. These branches are merged into a single convolution kernel post-training, allowing for efficient inference. RepLKNet Ding et al. (2022b) and SLaK Liu et al. (2022a) extend this approach to large 2D kernels (e.g., $31 \times 31$ and $51 \times 51$), increasing the receptive field while maintaining tractable inference cost via kernel decomposition or sparse groups. A complementary line of work focuses on gradient re-parameterization instead of modifying model weights directly. RepOptimizer Ding et al. (2022a), for example, modifies the back-propagation process by applying learnable scaling to gradient updates, enabling effective training of plain CNNs. These techniques reduce reliance on complex architectural design and have been shown to match or exceed the performance of more intricate networks. While much of the re-parameterization research has focused on 2D natural images, extending these methods to 3D medical imaging presents unique challenges. Volumetric kernels require significantly more parameters, and naive kernel expansion leads to high computational costs and optimization instability. 3D RepUX-Net Lee et al. (2023) demonstrate the inital attempt of adapting weight re-parameterization to 3D medical imaging and scale large depthwise kernels with fixed prior context, but still lacks of flexibility on adapting dynamic variation in fine-grained semantics for learning convergence. To bridge this gap, there is growing interest in using spatial priors or effective receptive field modeling to guide re-parameterization for large kernel learning in the 3D setting.

## 3 Rep3D

Rep3D rethinks the training dynamics of large-kernel convolution by explicitly embedding spatial bias, derived from effective receptive fields (ERFs), into the optimization process. Motivated by structural reparameterization (SR) and the distinctive gradient behavior observed in ERFs, Rep3D introduces a low-rank, learnable reparameterization that adapts element-wise update behavior across the kernel. We first derive the theoretical equivalence between parallel convolution branches and their single-operator counterparts, showing that a "large + small" convolution block (as in RepLKNet Ding et al. (2022b)) implicitly assigns spatially varying learning rates. We then translate this insight into a unified formulation and construct a lightweight generator that outputs a convergence-aware modulation mask, as shown in Figure 1. The output modulated mask models fine-grained learning dynamics during training, improving both scalability and performance in 3D tasks with large kernel convolution.

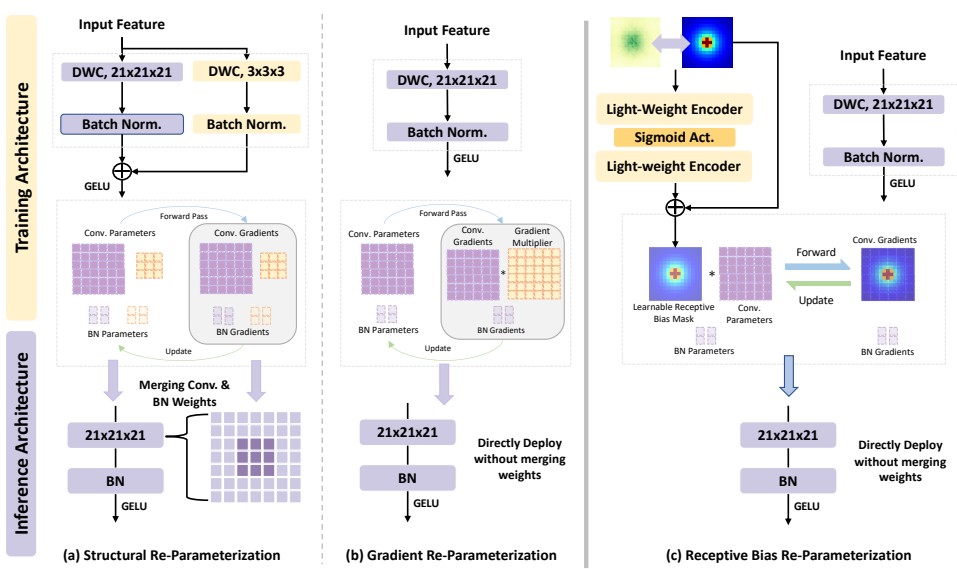

Figure 2: In contrast to (a) structural or (b) gradient-based re-parameterization, Rep3D introduces a novel re-parameterization strategy that injects a learnable spatial bias into large kernel convolutions for optimization. During training, a lightweight generator network produces a modulation mask conditioned on a distance-based prior, which adaptively scales gradient updates across the kernel. This enables spatially-aware learning dynamics that reflect local-to-global variations in the effective receptive field (ERF).

## 3.1 VARIABLE LEARNING CONVERGENCE IN PARALLEL BRANCH

From previous studies that is shown in Figure 2a and 2b, the learning convergence of the large kernel convolution can be improved by either adding up the encoded outputs of parallel branches weighted by diverse scales with SR (RepLKNet Ding et al. (2022b)) or performing Gradient Reparameterization (GR) by multiplying with constant values (RepOptimizer Ding et al. (2022a)) in a Single Operator (SO). Inspired by the concepts of structural re-parameterization (SR) and gradient re-parameterization, we extend the theoretical derivation from RepOptimizer and observe the variable learning rate across branches. We begin by analyzing the CSLA block, a basic two-branch design used in structural reparameterization (SR)-based networks (e.g., RepLKNet Ding et al. (2022b)). Let $X$ denote the input feature map, and let $W_L, W_S$ be large and small 3D convolution kernels, scaled by fixed positive scalars $\alpha_L$ and $\alpha_S$, respectively. The output of the CSLA module is:

$$Y_{\text{CSLA}} = \alpha_L(X * W_L) + \alpha_S(X * W_S), \tag{1}$$

where $*$ denotes 3D convolution. To unify the branches into a single equivalent convolution for efficient inference, we define a single-operator (SO) form:

$$Y_{\text{SO}} = X * W', \tag{2}$$

where the equivalent kernel $W'$ is a linear combination of the two branches:

$$W' = \alpha_L W_L + \alpha_S W_S. \tag{3}$$

During training with first-order optimization (i.e. SGD, AdamW) and step size $\lambda$, we apply the stochastic gradient descent rule and update the gradients for the parallel branches as follow:

$$W'_{t+1} = W'_t - \lambda \frac{\partial \mathcal{L}}{\partial W'_t}. \tag{4}$$

As the parallel branch architecture updates $W_L$ and $W_S$ independently:

$$W_{L(t+1)} = W_{L(t)} - \lambda_L \frac{\partial \mathcal{L}}{\partial W_{L(t)}}, \quad W_{S(t+1)} = W_{S(t)} - \lambda_S \frac{\partial \mathcal{L}}{\partial W_{S(t)}}. \tag{5}$$

where $\lambda_L$ and $\lambda_S$ are the learning rate for corresponding branch respectively. Substituting these into the equivalent kernel formulation yields:

$$W'_{t+1} = \alpha_L W_{L(t+1)} + \alpha_S W_{S(t+1)} \tag{6}$$

$$= \alpha_L \left( W_{L(t)} - \lambda_L \frac{\partial \mathcal{L}}{\partial W_{L(t)}} \right) + \alpha_S \left( W_{S(t)} - \lambda_S \frac{\partial \mathcal{L}}{\partial W_{S(t)}} \right) \tag{7}$$

$$= \alpha_L W_{L(t)} + \alpha_S W_{S(t)} - \lambda_L \alpha_L \frac{\partial \mathcal{L}}{\partial W_{L(t)}} - \lambda_S \alpha_S \frac{\partial \mathcal{L}}{\partial W_{S(t)}} \tag{8}$$

$$= W'_t - \lambda_L \alpha_L \frac{\partial \mathcal{L}}{\partial W_{L(t)}} - \lambda_S \alpha_S \frac{\partial \mathcal{L}}{\partial W_{S(t)}} \tag{9}$$

From the equation 9, we observe that each branch can be optimized differently with different learning rates toward each kernel and derive with two distinctive scenarios as follow:

$$W'_{t+1} = \begin{cases} W'_t - \lambda \left( \alpha_L \frac{\partial \mathcal{L}}{\partial W_{L(t)}} + \alpha_S \frac{\partial \mathcal{L}}{\partial W_{S(t)}} \right), & \text{if} \quad \lambda_L = \lambda_S \\ W'_t - \lambda_L \alpha_L \frac{\partial \mathcal{L}}{\partial W_{L(t)}} - \lambda_S \alpha_S \frac{\partial \mathcal{L}}{\partial W_{S(t)}}, & \text{if} \quad \lambda_L \neq \lambda_S \end{cases} \tag{10}$$

By the chain rule, we further derive:

$$\frac{\partial \mathcal{L}}{\partial W_{L(t)}} = \frac{\partial \mathcal{L}}{\partial Y_{\text{CSLA}}} \cdot \frac{\partial Y_{\text{CSLA}}}{\partial W_{L(t)}} = \alpha_L \cdot \frac{\partial \mathcal{L}}{\partial Y_{\text{CSLA}}} \cdot \frac{\partial (X * W_L)}{\partial W_{L(t)}}, \tag{11}$$

$$\frac{\partial \mathcal{L}}{\partial W_{S(t)}} = \frac{\partial \mathcal{L}}{\partial Y_{\text{CSLA}}} \cdot \frac{\partial Y_{\text{CSLA}}}{\partial W_{S(t)}} = \alpha_S \cdot \frac{\partial \mathcal{L}}{\partial Y_{\text{CSLA}}} \cdot \frac{\partial (X * W_S)}{\partial W_{S(t)}}. \tag{12}$$

To validate the above theoretical derivation, we perform ablation studies and found out that variable learning rate of each branch (i.e. $\lambda_S = 0.0006, \lambda_L = 0.0002$) demonstrates the best performance with stochastic gradient descent. Since $W_S$ has a smaller receptive field than $W_L$, and $W_S$ primarily contributes to the central region of the equivalent kernel $W'$, we argue that:

- **Central region of $W'$:** receives gradient contributions from both $W_L$ and $W_S$, resulting in faster convergence and stronger local learning.

- **Peripheral region of $W'$:** receives gradients only from $W_L$, leading to slower convergence but maintaining global contextual awareness.

Both the coefficients $\alpha_L$ and $\alpha_S$ modulate spatially distinct regions (large kernel contributions dominate the periphery, small kernel contributions dominate the central region), the two–branch block demonstrates a learning-rate field of:

$$\lambda_{\text{eff}}(\Delta x) = \begin{cases} \alpha_L \, \lambda_L, & \text{peripheral offsets,} \\ \alpha_L \, \lambda_L + \alpha_S \, \lambda_S, & \text{central offsets,} \end{cases} \tag{13}$$

where $\lambda_{\text{eff}}$ is the effective element-wise learning rate inherited from the two branch-specific updates.

## 3.2 Low-Rank Receptive Bias Modeling (LRBM)

As the above theory further validates the correlation between variable learning with the local-to-global gradient dynamics in ERF, we argue that such receptive bias can enhance the efficiency of learning large convolution kernels. We model the diffusion behavior of ERF with a reciprocal distance decay function $f_d$ and generate a prior mapping $P \in \mathbb{R}^{C \times 1 \times K \times K \times K}$ for weight re-parameterization as follow:

$$f_d(x, y, z, c) = \sqrt{(x - c)^2 + (y - c)^2 + (z - c)^2}$$

$$P = \frac{\beta}{d(x_k, y_k, z_k, c) + \beta} \tag{14}$$

where $k$ and $c$ are the element and central index of the kernel weight, $\beta$ is a learnable parameter to control the weight distribution of the distance mapping and initialize as 0. However, such a fixed prior mapping lacks of flexibility to adapt the weighting importance dynamically across the fine-grained

semantic variations in medical imaging. To address this, we propose to adapt learnable spatial bias by co-training a light-weight 2-layer generator network $f_\theta : \mathbb{R}^{C \times 1 \times K \times K \times K} \to \mathbb{R}^{C \times 1 \times K \times K \times K}$. We generate an adaptive mask $M$ for depthwise convolution kernels with low computation cost as follows:

$$M = P + f_\theta(P) \tag{15}$$

$$f_\theta(P) = \text{Norm}_2(\text{DConv}_2\left(\sigma\left(\text{Norm}_1(\text{DConv}_1(P))\right)\right)) \tag{16}$$

where $\text{DConv}_1$ and $\text{DConv}_2$ are 3D depthwise convolutions with kernel size of 7 and padding of 3, both $\text{Norm}_1$ $\text{Norm}_2$ are the layer normalizations, and $\sigma$ is a non-linear sigmoid activation to ensure all scaling value between 0 and 1. Such learnable function aims to capture the dynamic weighting of each kernel elements across local to global, while preserving computational efficiency. The resulting modulation mask $M$ is then used to reparameterize the kernel weights:

$$W_{\text{eff}} = W \odot M \tag{17}$$

where $W$ is the original convolution kernel and $\odot$ denotes element-wise multiplication. Importantly, the mask is applied during training only and the learned generator can be removed during inference for efficiency.

### 3.3 Network Architecture

The overall network architecture to validate Rep3D builds upon the encoder–decoder structure of 3D UX-Net Lee et al. (2022), which processes volumetric data through hierarchical resolution stages with skip connections to preserve fine-grained spatial features. Unlike prior transformer-based models or heavily modular CNNs, our design favors plain convolution blocks to minimize computational burden while preserving capacity for large-scale context modeling. Following the insights from prior work Ding et al. (2022b), we adopt a $21 \times 21 \times 21$ depthwise convolution (DWC-21) as the kernel backbone, which we empirically identify as the best trade-off between expressiveness and efficiency in 3D. Each encoder block consists of batch normalization, followed by the depthwise convolution and GeLU activation. The feature propagation from layer $\ell - 1$ to $\ell$ and then to $\ell + 1$ is defined as:

$$\hat{z}_\ell = \text{GELU}\left(\text{DWC-21}(\text{BN}(z_{\ell-1}))\right), \quad \hat{z}_{\ell+1} = \text{GELU}\left(\text{DWC-21}(\text{BN}(\hat{z}_\ell))\right) \tag{18}$$

where $z_{\ell-1}$ is the input from the previous layer, $\hat{z}\ell$ and $\hat{z}\ell + 1$ are intermediate representations, BN denotes batch normalization, and DWC-21 represents depthwise convolution with a $21^3$ kernel. This architectural choice allows the network to efficiently encode both local and global context, while enabling seamless integration of our re-parameterized learning framework (as detailed in section 3.1 and 3.2). The simplicity of the block ensures compatibility with the spatial modulation mask described in the next section and avoids unnecessary overhead during both training and inference.

## 4 Experimental Setup

**Datasets and Implementation Details.** We evaluate Rep3D on four publicly available volumetric segmentation datasets, covering a wide range of anatomical structures across different spatial scales, from large organs (e.g., liver, stomach) to smaller and more challenging targets (e.g., tumors, vessels). We report results using the Dice Similarity Coefficient (DSC) as the primary evaluation metric, quantifying spatial overlap between predicted segmentations and ground truth labels. Additional details, including dataset resolution normalization, voxel spacing, and pre-processing pipelines and experimental details are provided in the appendix.

## 5 Results

### 5.1 Evaluation on Tissue & Tumor Segmentation

To assess the generalization and scalability of Rep3D across diverse anatomical structures and clinical targets, we evaluate performance on three representative volumetric segmentation tasks using the KiTS, MSD Pancreas, and MSD Hepatic Vessel datasets. As shown in Table 3, Rep3D achieves state-of-the-art performance across all settings, consistently outperforming both convolution- and transformer-based baselines. On the KiTS dataset, which includes kidney, tumor, and cyst

Table 1: Comparison of SOTA approaches on the three different testing datasets. (*: $p < 0.01$, with Paired Wilcoxon signed-rank test to all baseline networks)

| Methods | #Params | FLOPs | KiTS | | | | MSD | | | | | |
|---|---|---|---|---|---|---|---|---|---|---|---|---|
| | | | Kidney | Tumor | Cyst | Mean | Pancreas | Tumor | Mean | Hepatic | Tumor | Mean |
| 3D U-Net Çiçek et al. (2016) | 4.81M | 135.9G | 0.918 | 0.657 | 0.361 | 0.645 | 0.711 | 0.584 | 0.648 | 0.569 | 0.609 | 0.589 |
| SegResNet Myronenko (2018) | 1.18M | 15.6G | 0.935 | 0.713 | 0.401 | 0.683 | 0.740 | 0.613 | 0.677 | 0.620 | 0.656 | 0.638 |
| RAP-Net Lee et al. (2021) | 38.2M | 101.2G | 0.931 | 0.710 | 0.427 | 0.689 | 0.742 | 0.621 | 0.682 | 0.610 | 0.643 | 0.627 |
| nn-UNet Isensee et al. (2021) | 31.2M | 743.3G | 0.943 | 0.732 | 0.443 | 0.706 | 0.775 | 0.630 | 0.703 | 0.623 | 0.695 | 0.660 |
| TransBTS Wang et al. (2021) | 31.6M | 110.3G | 0.932 | 0.691 | 0.384 | 0.669 | 0.749 | 0.610 | 0.679 | 0.589 | 0.636 | 0.613 |
| UNETR Hatamizadeh et al. (2022b) | 92.8M | 82.5G | 0.921 | 0.669 | 0.354 | 0.648 | 0.735 | 0.598 | 0.667 | 0.567 | 0.612 | 0.590 |
| nnFormer Zhou et al. (2021) | 149.3M | 213.0G | 0.930 | 0.687 | 0.376 | 0.664 | 0.769 | 0.603 | 0.686 | 0.591 | 0.635 | 0.613 |
| SwinUNETR Hatamizadeh et al. (2022a) | 62.2M | 328.1G | 0.939 | 0.702 | 0.400 | 0.680 | 0.785 | 0.632 | 0.708 | 0.622 | 0.647 | 0.635 |
| 3D UX-Net (k=7) Lee et al. (2022) | 53.0M | 639.4G | 0.942 | 0.724 | 0.425 | 0.697 | 0.737 | 0.614 | 0.676 | 0.625 | 0.678 | 0.652 |
| UNesT-B Yu et al. (2023) | 87.2M | 258.4G | 0.943 | 0.746 | 0.451 | 0.710 | 0.778 | 0.601 | 0.690 | 0.611 | 0.645 | 0.640 |
| Rep3D (Fixed Prior) | 65.8M | 757.4G | 0.950 | 0.757 | 0.473 | 0.727 | 0.789 | 0.640 | 0.715 | 0.635 | 0.681 | 0.658 |
| Rep3D | 66.0M | 757.6G | **0.955** | **0.763** | **0.490** | **0.736*** | **0.793** | **0.653** | **0.723*** | **0.650** | **0.697** | **0.674*** |

Table 2: Evaluations on the AMOS testing split in different scenarios.(*: $p < 0.01$, with Paired Wilcoxon signed-rank test to all baseline networks)

| | | | | | | | AMOS CT (Train From Scratch Scenario) | | | | | | | | | |
|---|---|---|---|---|---|---|---|---|---|---|---|---|---|---|---|---|
| Methods | Spleen | R. Kid | L. Kid | Gall. | Eso. | Liver | Stom. | Aorta | IVC | Panc. | RAG | LAG | Duo. | Blad. | Pros. | Avg |
| nn-UNet (350 Epochs) | 0.951 | 0.919 | 0.930 | 0.845 | 0.797 | 0.975 | 0.863 | 0.941 | 0.898 | 0.813 | 0.730 | 0.677 | 0.772 | 0.797 | 0.815 | 0.850 |
| nn-UNet (1000 Epochs) | 0.967 | 0.958 | 0.945 | 0.890 | 0.818 | 0.979 | 0.914 | 0.953 | 0.920 | 0.824 | 0.799 | 0.743 | 0.823 | 0.900 | 0.867 | 0.887 |
| TransBTS | 0.930 | 0.921 | 0.909 | 0.798 | 0.722 | 0.966 | 0.801 | 0.900 | 0.820 | 0.702 | 0.641 | 0.550 | 0.684 | 0.730 | 0.679 | 0.783 |
| UNETR | 0.925 | 0.923 | 0.903 | 0.777 | 0.701 | 0.964 | 0.759 | 0.887 | 0.821 | 0.687 | 0.688 | 0.543 | 0.629 | 0.710 | 0.707 | 0.740 |
| nnFormer | 0.932 | 0.928 | 0.914 | 0.831 | 0.743 | 0.968 | 0.820 | 0.905 | 0.838 | 0.725 | 0.678 | 0.578 | 0.677 | 0.737 | 0.596 | 0.785 |
| SwinUNETR | 0.956 | 0.957 | 0.949 | 0.891 | 0.820 | 0.978 | 0.880 | 0.939 | 0.894 | 0.818 | 0.800 | 0.730 | 0.803 | 0.849 | 0.819 | 0.871 |
| 3D UX-Net (k=7) | 0.966 | 0.959 | 0.951 | 0.903 | 0.833 | 0.980 | 0.910 | 0.950 | 0.913 | 0.830 | 0.805 | 0.756 | 0.846 | 0.897 | 0.863 | 0.890 |
| 3D UX-Net (k=21) | 0.963 | 0.959 | 0.953 | 0.921 | 0.848 | 0.981 | 0.903 | 0.953 | 0.910 | 0.828 | 0.815 | 0.754 | 0.824 | 0.900 | 0.878 | 0.891 |
| UNesT-B | 0.966 | 0.961 | 0.956 | 0.903 | 0.840 | 0.980 | 0.914 | 0.947 | 0.912 | 0.838 | 0.803 | 0.758 | 0.846 | 0.895 | 0.854 | 0.891 |
| RepOptimizer | 0.968 | 0.964 | 0.953 | 0.903 | 0.857 | 0.981 | 0.915 | 0.950 | 0.915 | 0.826 | 0.802 | 0.756 | 0.813 | 0.906 | 0.867 | 0.892 |
| Rep3D (Fixed Prior) | 0.972 | 0.963 | 0.964 | 0.911 | 0.861 | 0.982 | 0.921 | 0.956 | 0.924 | 0.837 | 0.818 | 0.777 | 0.831 | 0.916 | 0.879 | 0.902 |
| Rep3D (LRBM) | **0.978** | **0.970** | **0.964** | **0.928** | **0.871** | **0.984** | **0.927** | **0.960** | **0.930** | **0.851** | **0.828** | **0.784** | **0.850** | **0.920** | **0.881** | **0.910*** |
| | | | | | | | AMOS MRI (Train From Scratch Scenario) | | | | | | | | | |
| Methods | Spleen | R. Kid | L. Kid | Gall. | Eso. | Liver | Stom. | Aorta | IVC | Panc. | RAG | LAG | Duo. | Blad. | Pros. | Avg |
| nn-UNet (350 Epochs) | 0.967 | 0.855 | 0.958 | 0.663 | 0.736 | 0.973 | 0.888 | 0.956 | 0.907 | 0.793 | 0.533 | 0.572 | 0.668 | - | - | 0.805 |
| nn-UNet (1000 Epochs) | 0.973 | 0.940 | 0.965 | **0.681** | 0.810 | 0.979 | **0.893** | 0.967 | **0.917** | 0.834 | 0.667 | 0.689 | 0.701 | - | - | 0.847 |
| TransBTS | 0.956 | 0.957 | 0.955 | 0.619 | 0.770 | 0.974 | 0.867 | 0.958 | 0.852 | 0.836 | 0.591 | 0.630 | 0.648 | - | - | 0.816 |
| UNETR | 0.942 | 0.956 | 0.930 | 0.552 | 0.741 | 0.967 | 0.836 | 0.947 | 0.829 | 0.815 | 0.564 | 0.621 | 0.624 | - | - | 0.794 |
| nnFormer | 0.949 | 0.952 | 0.950 | 0.601 | 0.758 | 0.972 | 0.859 | 0.960 | 0.843 | 0.832 | 0.569 | 0.618 | 0.637 | - | - | 0.808 |
| SwinUNETR | 0.972 | 0.961 | 0.961 | 0.649 | 0.814 | 0.978 | 0.889 | 0.961 | 0.862 | 0.854 | 0.659 | 0.649 | 0.664 | - | - | 0.836 |
| 3D UX-Net (k=7) | 0.971 | 0.965 | 0.966 | 0.603 | 0.828 | 0.978 | 0.869 | 0.962 | 0.878 | 0.837 | 0.696 | 0.689 | 0.696 | - | - | 0.841 |
| 3D UX-Net (k=21) | 0.968 | 0.962 | 0.967 | 0.610 | 0.830 | 0.977 | 0.858 | 0.954 | 0.880 | 0.829 | 0.701 | 0.697 | 0.700 | - | - | 0.840 |
| UNesT-B | 0.971 | 0.965 | 0.967 | 0.615 | 0.831 | 0.980 | 0.865 | 0.949 | 0.883 | 0.845 | 0.691 | 0.700 | 0.697 | - | - | 0.843 |
| RepOptimizer | 0.970 | 0.967 | 0.971 | 0.635 | 0.823 | 0.978 | 0.875 | 0.963 | 0.882 | 0.850 | 0.689 | 0.691 | 0.711 | - | - | 0.847 |
| Rep3D (Fixed Prior) | 0.972 | 0.965 | 0.970 | 0.644 | 0.838 | 0.980 | 0.883 | 0.965 | 0.893 | 0.861 | 0.714 | 0.701 | 0.725 | - | - | 0.855 |
| Rep3D (LRBM) | **0.975** | **0.969** | **0.975** | 0.657 | **0.845** | **0.984** | 0.891 | **0.970** | 0.901 | **0.879** | **0.718** | **0.721** | **0.750** | - | - | **0.864*** |

segmentation, Rep3D achieves the highest average Dice score of 0.736, with strong individual scores of 0.955 (kidney), 0.763 (tumor), and 0.490 (cyst). Notably, Rep3D improves tumor segmentation performance by 2.28% Dice over UNesT-B and 5.39% Dice over 3D UX-Net, demonstrating its ability to adapt to complex local variations in pathological regions. On the MSD Pancreas task, which is particularly challenging due to the pancreas's low contrast and irregular boundaries, Rep3D sets a new benchmark with an average Dice score of 0.723, outperforming SwinUNETR (0.708), nnUNet (0.703), and UNesT-B (0.690). Tumor segmentation also benefits from our re-parameterization design, improving by 3.32% Dice compared to 3D UX-Net and 2,03% Dice compared to the fixed-prior variant. On the MSD Hepatic Vessel dataset, Rep3D continues to lead with a mean Dice of 0.674, outperforming the previous best model (UNesT-B, 0.640) and demonstrating superior vessel and tumor localization. The results also highlight the effectiveness of Rep3D's spatially adaptive learning dynamics, especially in sparse and small-structure segmentation where traditional large-kernel convolutions or global self-attention tend to underperform.

## 5.2 EVALUATION ON MULTI-ORGAN SEGMENTATION

Beyond the ability to segment anatomical structures across scales, we furtehr evaluate Rep3D on the AMOS benchmark under the "train-from-scratch" setting for both CT and MRI modalities. On AMOS-CT, Rep3D achieves the best performance across all 15 evaluated anatomical structures, surpassing strong baselines including SwinUNETR, UNesT, and 3D UX-Net. Notably, Rep3D outperforms UNesT-B by 2.13% and RepOptimizer by 2.02% of average Dice score, while operating

Table 3: Ablation Studies on bias generator's convolutional layers and LRBM 3D adaptability

| Methods | Spleen | R. Kid | L. Kid | Gall. | Eso. | Liver | Stom. | Aorta | IVC | Panc. | RAG | LAG | Duo. | Blad. | Pros. | Avg |
|---|---|---|---|---|---|---|---|---|---|---|---|---|---|---|---|---|
| Kernel=$1 \times 1 \times 1$ | 0.972 | 0.968 | **0.965** | 0.926 | 0.863 | 0.984 | 0.917 | 0.956 | 0.922 | 0.851 | 0.816 | 0.779 | **0.863** | 0.912 | **0.894** | 0.905 |
| Kernel=$3 \times 3 \times 3$ | 0.970 | 0.966 | 0.960 | **0.930** | 0.863 | 0.984 | **0.935** | 0.958 | 0.924 | **0.859** | 0.827 | 0.758 | 0.862 | 0.908 | 0.892 | 0.906 |
| Kernel=$5 \times 5 \times 5$ | 0.974 | 0.967 | 0.964 | 0.925 | 0.833 | 0.984 | 0.924 | 0.956 | 0.910 | 0.850 | **0.829** | **0.786** | 0.843 | **0.921** | 0.884 | 0.903 |
| Kernel=$7 \times 7 \times 7$ | **0.978** | **0.970** | 0.964 | 0.928 | **0.871** | 0.984 | 0.927 | **0.960** | **0.930** | 0.851 | 0.828 | 0.784 | 0.850 | 0.920 | 0.881 | **0.910** |
| 3D UX-Net (k=7) | 0.966 | 0.959 | 0.951 | 0.903 | 0.833 | 0.980 | 0.910 | 0.950 | 0.913 | 0.830 | 0.805 | 0.756 | 0.846 | 0.897 | 0.863 | 0.890 |
| 3D UX-Net + LRBM | 0.968 | 0.963 | 0.952 | 0.911 | 0.841 | 0.981 | 0.915 | 0.959 | 0.920 | 0.835 | 0.811 | 0.770 | 0.851 | 0.901 | 0.872 | 0.897 |

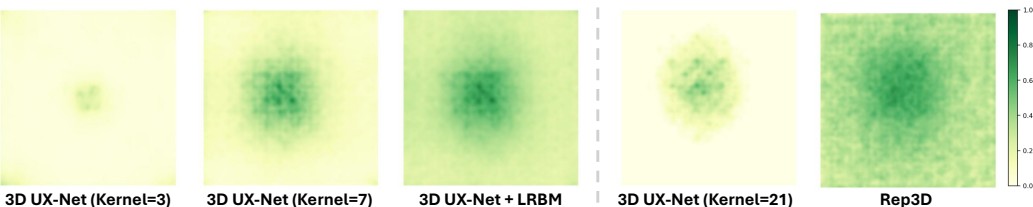

**3D UX-Net (Kernel=3)**  **3D UX-Net (Kernel=7)**  **3D UX-Net + LRBM**  **3D UX-Net (Kernel=21)**  **Rep3D**

Figure 3: As kernel size increases, depthwise convolutions in 3D UX-Net exhibit increasingly diffuse ERFs, gradually expanding the gradient dynamics from local to broader spatial regions. Incorporating LRBM further enhances weighting toward global areas by modulating the spatial contribution of distant elements. In contrast, Rep3D produces a well-distributed ERF that preserves strong central activation while extending contextual influence across the full kernel.

with fewer parameters than UNesT. On AMOS-MRI, a more challenging modality due to the variable range of contrast intensity and anatomical ambiguity, Rep3D maintains its superior performance, achieving an average Dice of 0.864, again outperforming all competing approaches. Compared to the best-performing transformer baseline (UNesT-B, 0.854) and convolutional baseline (3D UX-Net (k=21), 0.840), Rep3D delivers consistent improvements across nearly all organ classes, particularly in difficult regions such as the pancreas, gallbladder, and adrenal glands. These gains underscore the effectiveness of our spatially adaptive re-parameterization strategy in enhancing convergence and feature expressivity without increasing model complexity.

## 5.3 ABLATION STUDIES

**Effect of Network Depth for LRBM.** To investigate the impact of architectural depth in the spatial modulation generator, we conduct an ablation study by varying the number of layers in the generator network used to produce the element-wise modulation mask in Rep3D (in supplementart material). Specifically, we compare shallow configurations (1-layer depthwise convolution) with deeper variants (2-layer and 3-layer depthwise convolution stacks), while keeping the total parameter count approximately constant. Our results show that the 2-layer design provides the best trade-off between representation flexibility and training stability. While the 1-layer generator lacks sufficient capacity to capture nuanced spatial priors, resulting in under-modulated gradient flow. The 3-layer version demonstrates a slight decrease of performance (from 0.910 to 0.899 Dice) and instability during training. This suggests that a lightweight, moderately deep generator is optimal for learning spatially adaptive convergence patterns without incurring additional complexity or over-parameterization.

**Effect of Kernel Size in Spatial Bias Modeling.** To further understand how kernel size affects segmentation performance across different anatomical structures, we analyzed organ-wise performance under varying kernel configurations: $1 \times 1 \times 1$, $3 \times 3 \times 3$, $5 \times 5 \times 5$, and $7 \times 7 \times 7$ used in Rep3D. All configurations share the same training protocol and re-parameterization setup, isolating the effect of kernel size alone. As shown in Table 3, the impact of kernel size varies across organs. While the $7 \times 7 \times 7$ kernel achieves the highest overall mean Dice score (0.910), smaller or boundary-sensitive organs (e.g., bladder, adrenal glands) benefit from small- or mid-size kernels such as $1 \times 1 \times 1$ or $5 \times 5 \times 5$. In contrast, large organs with strong spatial continuity (e.g., liver, spleen, aorta) show clear improvements with larger receptive fields. These results suggest that optimal kernel size is organ-dependent, influenced by factors such as spatial extent, anatomical context, and structural complexity. The superior performance of the $7 \times 7 \times 7$ variant reflects its ability to balance local detail and global context.

**Effect of LRBM towards Other Network Architectures.** To isolate the contribution of our proposed Low-Rank Bias Modeling (LRBM) module, we integrate it into a standard 3D UX-Net architecture (with fixed $7 \times 7 \times 7$ kernels) and compare its performance to the original baseline. As reported in Table 3, incorporating LRBM improves the average Dice score from 0.890 to 0.897, with consistent gains across multiple organs including pancreas, bladder, and adrenal glands. While the improvement may appear modest in aggregate, it is particularly noteworthy in anatomically challenging regions where gradient convergence is often unstable. For example, performance on the left adrenal gland increases from 0.756 to 0.770, and the duodenum improves from 0.846 to 0.851, suggesting that the learnable spatial bias improves optimization dynamics in fine-scale structures. These results confirm that our LRBM module offers a generalizable and plug-and-play mechanism for enhancing 3D segmentation backbones, even outside the full Rep3D framework.

## 6 DISCUSSIONS & LIMITATIONS

In this work, we introduced Rep3D, a re-parameterization framework that explicitly models spatial convergence dynamics in large kernel 3D convolutions. By linking effective receptive field (ERF) behavior with first-order optimization theory, we demonstrated that large convolution kernels naturally exhibit non-uniform learning dynamics, where central elements converge faster than peripheral ones. To address this, Rep3D integrates a learnable spatial prior via low-rank modulation, allowing the optimizer to differentially emphasize kernel regions with the distinctive characteristics of ERF during training. Our experiments across five diverse 3D segmentation benchmarks, confirm that Rep3D consistently improves performance over both transformer-based and convolution-based SOTA approaches, while maintaining a plain and efficient encoder design. The success of Rep3D reinforces several broader insights. First, spatially adaptive optimization is a promising direction for bridging inductive biases in CNNs with the dynamic learning capacity of attention-based models. Second, incorporating explicit ERF modeling into kernel design enables more efficient parameter usage, particularly in data-limited medical imaging scenarios. Moreover, our framework enhance network interpretability: the modulation masks can be visualized and aligned with ERF pattern as demonstrated in Figure 3, offering insights into how spatial understanding guides the learning of convolution kernels.

While Rep3D demonstrates strong empirical performance across diverse 3D medical segmentation tasks, several limitations remain. First, although our learnable modulation mechanism introduces minimal architectural overhead, the training cost associated with large 3D kernels (e.g., $21 \times 21 \times 21$) remains nontrivial, particularly in memory-constrained GPU environments. Unlike 2D convolution kernels (i.e. MegEngine packages for 2D depthwise kernels), limited packages and approaches has been proposed to optimize the large kernel mechanism in 3D. This limits the batch size and input resolution during training, which can affect convergence and generalization. Future work could explore progressive training strategies, multi-resolution optimization, or low-resolution proxy supervision to alleviate this constraint while maintaining segmentation fidelity. Second, while our distance decay prior effectively guides spatial re-parameterization, its performance is inherently tied to the input volume resolution. In our experiments, we downsample 3D volumes to specific resolution (e.g., $1.5 \times 1.5 \times 2.0$ mm) to balance computation and efficency. However, we observe saturation effects when training at higher resolutions, where further improvements in image quality do not yield proportional gains in segmentation accuracy. This may be due to the spatial prior losing precision at finer scales. Adapting fine-grained spatial learnable prior could be another potential direction for future work.

## 7 CONCLUSION

In this paper, we introduced Rep3D, a receptive-biased re-parameterization framework for large kernel 3D convolutions. By modeling effective receptive field (ERF) behavior as a learnable spatial prior, Rep3D enables adaptive element-wise learning dynamics during training, bridging the gap between convolutional inductive bias and optimization-aware design. Implemented via a lightweight modulation network, our approach avoids complex multi-branch architectures while improving training efficiency and segmentation accuracy. Extensive experiments across five volumetric medical imaging benchmarks demonstrate consistent improvements over SOTA transformer and CNN approaches, establishing Rep3D as a scalable and effective solution for 3D medical image analysis.

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

# A  APPENDIX

## A.1  REP3D MOTIVATIONS FROM EXPERIMENTS: NAIVELY SCALING 3D CONVOLUTION KERNELS

Recent advances in medical image segmentation have highlighted the importance of large receptive fields for capturing long-range spatial dependencies in volumetric data. Motivated by this, there has been a growing trend toward enlarging convolutional kernel sizes in 3D CNN architectures, such as 3D UX-Net Lee et al. (2022) and RepUX-Net Lee et al. (2023), which attempt to mimic the global context modeling capabilities of transformers while retaining the inductive biases of CNNs.

However, the straightforward enlargement of kernel size introduces several practical and theoretical challenges:

- **Optimization Instability:** Large convolutional kernels suffer from slow or uneven convergence, particularly in the outer kernel regions, which are rarely activated in early training. This leads to ineffective utilization of capacity and suboptimal learning behavior.

- **Degrading Performance:** Empirically, simply increasing the kernel size does not guarantee improved performance. Beyond a certain scale, performance tends to saturate or even degrade.

- **Inefficient Parameter Usage:** Naïve kernel scaling introduces a quadratic (especially in 3D) growth in the number of parameters, making training inefficient and difficult to regularize.

To illustrate these effects empirically, we conduct a systematic ablation study on the AMOS dataset using the 3D UX-Net encoder as a backbone. 3D UX-Net is an ideal starting point because it already leverages larger kernels ($7 \times 7 \times 7$) in its baseline. We vary the convolutional kernel size from $3 \times 3 \times 3$ to $21 \times 21 \times 21$, keeping all other architectural and training settings fixed. The results, shown in Table 4 , confirm our hypothesis.

Table 4: Impact of kernel size on segmentation performance using 3D UX-Net encoder on the AMOS dataset.

| Kernel Size | Avg. Dice Score |
|---|---|
| $3 \times 3 \times 3$ | 0.881 |
| $5 \times 5 \times 5$ | 0.885 |
| $7 \times 7 \times 7$ | 0.890 |
| $9 \times 9 \times 9$ | 0.891 |
| $11 \times 11 \times 11$ | 0.893 |
| $13 \times 13 \times 13$ | 0.894 |
| $15 \times 15 \times 15$ | **0.895** |
| $17 \times 17 \times 17$ | 0.893 |
| $19 \times 19 \times 19$ | 0.893 |
| $21 \times 21 \times 21$ | 0.891 |

As seen in Table 4, performance initially improves with increasing kernel size, peaking at $15 \times 15 \times 15$. However, further enlargements yield diminishing or even negative returns, despite increased computational cost. These findings reveal a key limitation of naïve kernel enlargement: Although it increases theoretical receptive field, it fails to translate into meaningful gains due to the lack of spatially adaptive optimization.

## A.2  THEORETICAL EXTENSION FROM SGD TO ADAM/ADAMW

While our primary derivation (Equation (13)) assumes a first-order optimizer with a fixed learning rate (e.g., SGD), the key insight of **Rep3D** that spatially distributed kernel elements receive distinct effective learning rates through a low-rank re-parameterization, remains valid under adaptive optimizers such as Adam and AdamW.

In Adam, the parameter update rule is defined as:

$$w_t = w_{t-1} - \eta \cdot \frac{m_t}{\sqrt{v_t} + \epsilon} \tag{19}$$

where $m_t$ and $v_t$ denote the first and second moment estimates of the gradient, respectively. When applied to our low-rank spatial re-parameterization, the convolutional kernel weights are split into two branches:

$$W = \lambda \cdot W_L + (1 - \lambda) \cdot W_S \tag{20}$$

Each branch is updated independently and maintains separate moment statistics, resulting in branch-specific normalized gradients:

$$\Delta W_L \propto \frac{m_t^L}{\sqrt{v_t^L} + \epsilon}, \ \Delta W_S \propto \frac{m_t^S}{\sqrt{v_t^S} + \epsilon} \tag{21}$$

Although the global learning rate $\eta$ and the modulation coefficient $\lambda$ are shared, the effective update magnitude for each spatial location differs depending on its branch contribution. Specifically, due to their differing receptive field properties, the outer kernel regions are primarily influenced by the large branch, while the central kernel positions are governed by the small branch. This design induces **position-sensitive learning dynamics** even when using global optimizers like Adam. The adaptive normalization in Adam (via the second moment estimate) amplifies or attenuates updates differently across spatial positions, preserving the spatial bias induced by our re-parameterization structure.

In summary, although adaptive optimizers modify how gradients are scaled, the core Rep3D mechanism, independent per-branch update trajectories and element-wise optimization guided by spatial priors, remains intact and effective. We will provide the full derivation and corresponding results with AdamW in the final version for completeness.

## A.3 DATA PREPROCESSING & TRAINING DETAILS

Table 5: Hyperparameters for direction training scenario on four public datasets

| Hyperparameters | Direct Training |
|---|---|
| Encoder Stage | 4 |
| Layer-wise Channel | $48, 96, 192, 384$ |
| Hidden Dimensions | 768 |
| Patch Size | $96 \times 96 \times 96$ |
| No. of Sub-volumes Cropped | 2 |
| Training Steps | 60000 |
| Batch Size | 2 |
| AdamW $\epsilon$ | $1e - 8$ |
| AdamW $\beta$ | $(0.9, 0.999)$ |
| Peak Learning Rate | $1e - 4$ |
| Learning Rate Scheduler | ReduceLROnPlateau |
| Factor & Patience | 0.9, 10 |
| Dropout | X |
| Weight Decay | 0.08 |
| Data Augmentation | Intensity Shift, Rotation, Scaling |
| Cropped Foreground | ✓ |
| Intensity Offset | 0.1 |
| Rotation Degree | $-30°$ to $+30°$ |
| Scaling Factor | x: 0.1, y: 0.1, z: 0.1 |

We apply hierarchical steps for data preprocessing: 1) intensity clipping is applied to further enhance the contrast of soft tissue (AMOS CT, KiTS, MSD Pancreas:{min:-175, max:250}; MSD Hepatic Vessel:{min:0, max:230}); AMOS MRI:{min:0, max:1000}. 2) Intensity normalization is performed after clipping for each volume and use min-max normalization: $(X - X_1)/(X_{99} - X_1)$ to normalize

the intensity value between 0 and 1, where $X_p$ denote as the $p_{th}$ percentile of intensity in $X$. We then perform downsampling to certain voxel spacing (i.e. AMOS CT, MSD hepatic vessels, MSD Pancreas and KiTS: $1.5 \times 1.5 \times 2.0$, AMOS MRI: $1.0 \times 1.0 \times 1.0$) randomly crop sub-volumes with size $96 \times 96 \times 96$ at the foreground and perform data augmentations, including rotations, intensity shifting, and scaling (scaling factor: 0.1). All training processes with Rep3D are optimized with either Stochastic Gradient Descent (SGD) or AdamW optimizer. We trained all models for 60000 steps using a learning rate of 0.0001 on an NVIDIA A100 GPU across all datasets. One epoch takes approximately about 9 minute for KiTS, 5 minutes for MSD Pancreas, 12 minutes for MSD hepatic vessels, 7 minutes for AMOS CT and 1 minute for AMOS MRI, respectively.

All experiments are conducted under a direct supervised learning setting. For the KiTS and MSD datasets, we employ a 5-fold cross-validation strategy using an 80%/10%/10% split for training, validation, and testing, respectively. For the AMOS dataset, we use a fixed single split with the same partitioning ratio. Details on training procedures and preprocessing protocols are provided in the supplementary material. Our proposed re-parameterization approach Rep3D, is benchmarked against both convolutional and transformer-based state-of-the-art (SOTA) methods for 3D medical image segmentation. For nnUNet Isensee et al. (2021), we evaluate performance across two different training schedules to account for fairness, since Rep3D is trained with 60,000 iterations (approximately equivalent to 350 epochs). Therefore, we have provided performance with partial scheduled (350 epochs) and full scheduled (1000 epochs) to demonstrate our model generalizability.

## A.4 DATASETS DETAILS

We have leverage four challenging public datasets across different scales: 1) AMOS22 (MICCAI 2022 Abdominal Multi-organ Segmentation Challenge) Ji et al. (2022): Comprises 200 multi-contrast abdominal CT scans with 15 organ-level anatomical labels and 33 MRI scans with 13 organ-level anatomical labels for comprehensive abdominal segmentation, 2) KiTS21 (MICCAI 2021 Kidney Tumor Segmentation Challenge) Heller et al. (2019): Includes 210 contrast-enhanced abdominal CT scans from the University of Minnesota Medical Center (2010–2018), with manual annotations for kidney, tumor, and cyst, 3) MSD Pancreas (Medical Segmentation Decathlon) Antonelli et al. (2022): Contains 282 abdominal contrast-enhanced CT scans annotated for both pancreas and pancreatic tumor segmentation, and 4) MSD Hepatic Vessel (Medical Segmentation Decathlon) Antonelli et al. (2022): Contains 303 abdominal CT scans annotated for hepatic vessel and associated tumor segmentation.

Table 6: Complete overview of Four public datasets

| Challenge | AMOS CT | AMOS MR | MSD Pancreas | MSD Hepatic Vessels | KiTS |
|---|---|---|---|---|---|
| Imaging Modality | Multi-Contrast CT | Multi-Contrast MRI | Venous CT | | Arterial CT |
| Anatomical Region | Abdomen | | Pancreas | Liver | Kidney |
| Sample Size | 200 | 33 | 282 | 303 | 300 |
| Anatomical Label | Spleen, Left & Right Kidney, Gall Bladder, Esophagus, Liver, Stomach, Aorta, Inferior Vena Cava (IVC) Pancreas, Left & Right Adrenal Gland (AG), Duodenum Bladder (CT only), Prostate/Uterus (CT only) | | Pancreas, Tumor | Hepatic Vessels, Tumor | Kidney, Tumor |
| Data Splits | 1-Fold (Internal) | | 5-Fold Cross-Validation | | |
| | Train: 160 / Validation: 20 / Test: 20 | Train: 22 / Validation: 4/ Test: 7 | Train: 225 / Validation: 27 / Testing: 30 | Training: 242, Validation: 30 / Testing: 31 | Training: 240, Validation: 30 / Testing: 30 |
| 5-Fold Ensembling | N/A | N/A | X | ✓ | X |

## A.5 NETWORK ARCHITECTURE

We adopt a 3D encoder-decoder architecture from both 3D UX-Net Lee et al. (2022) and SwinUNETR Hatamizadeh et al. (2022a) as the backbone of Rep3D. Instead of using encoder block with feed forward layer, we simply using a plain convolutional design with depthwise separable convolutions in parallel with LRBM. The encoder consists of 4 hierarchical stages with increasing feature dimensions and depthwise convolutions of large kernel size ($21 \times 21 \times 21$), followed by a symmetric decoder for volumetric segmentation. The encoder includes:

- An initial input projection block with a $7 \times 7 \times 7$ convolution (stride 2, padding 3) followed by a residual block with two $3 \times 3 \times 3$ convolutions and GELU activations.
- Stage 1: 2 Rep3D blocks with 48 channels followed by a strided $2 \times 2 \times 2$ convolution for downsampling.
- Stage 2: 2 Rep3D blocks with 96 channels, followed by a strided $2 \times 2 \times 2$ convolution for downsampling.

- Stage 3: 2 Rep3D blocks with 192 channels, followed by a strided $2 \times 2 \times 2$ convolution for downsampling.
- Stage 4: 2 Rep3D blocks with 384 channels, followed by a strided $2 \times 2 \times 2$ convolution for downsampling.

Each stage modulates large kernel weights using a learnable re-parameterization mask computed via a lightweight 2-layer generator network within each Rep3D block. For each Rep3D block, it includes:

- A single depthwise 3D convolution with a large kernel size of $21 \times 21 \times 21$ and padding size of 10, followed by a layer normalization and a GELU activation.
- A 2-stage lightweight generator network including:
  - First layer: a depthwise $7 \times 7 \times 7$ convolution followed by layer normalization and a sigmoid activation.
  - Second layer: another depthwise $7 \times 7 \times 7$ convolution followed by layer normalization.

The decoder mirrors the encoder and consists of:

- 4 upsampling modules (UnetrUpBlock from MONAI), each with a transpose convolution (stride 2), skip connection, and a residual block with two $3 \times 3 \times 3$ convolutions and GELU activations.
- 1 output projection block (UnetOutBlock from MONAI) consisting of a $1 \times 1 \times 1$ convolution to map to the number of target classes.

### A.6 TRAINING EFFICIENCY COMPARISON

To further validate our claims around improved convergence behavior and training efficiency, we conducted an additional ablation study focusing on runtime and performance dynamics across different configurations of Rep3D on the AMOS dataset. While our primary goal is to improve segmentation accuracy and spatial convergence, it is equally important that such gains are achieved with minimal training overhead. In this study, we compare three architectural variants:

(a) **Vanilla Rep:** Rep3D with parallel convolutional branches, but without any spatial prior modulation.

(b) **Fixed Prior:** Rep3D with a non-learnable reciprocal distance mask acting as a fixed spatial prior.

(c) **Full LRBM:** Rep3D with a learnable low-rank bias module (LRBM), modulating the spatial prior adaptively via a generator network.

All models were trained under identical conditions: using a single NVIDIA A100 GPU, batch size of 2, and AdamW optimizer. We report the validation Dice scores at key training checkpoints (10k, 20k, 40k, and 60k iterations), as well as the total training time to convergence.

Table 7: Training efficiency and convergence comparison of different Rep3D variants on AMOS dataset.

| Method | Time (hrs) | 10k Iter | 20k Iter | 40k Iter | 60k Iter |
|---|---|---|---|---|---|
| Vanilla Rep | 17.3 | 0.853 | 0.868 | 0.886 | 0.892 |
| Fixed Prior | 15.5 | 0.864 | 0.875 | 0.892 | 0.902 |
| Full LRBM | 17.5 | **0.871** | **0.885** | **0.897** | **0.910** |

As observed in Table 7, the full LRBM variant consistently achieves the best segmentation accuracy at every checkpoint, demonstrating accelerated convergence. The introduction of the learnable low-rank generator yields a modest increase in training time ($+0.2$ hours compared to Vanilla Rep), but this is substantially outweighed by the observed performance gains. The Fixed Prior variant also performs competitively, suggesting the benefit of incorporating even a static spatial prior.

## A.7 VALIDATION EXPERIMENTS ON VARIABLE BRANCH LEARNING RATE

Table 8: Quantitative Evaluation on Variable Learning Rates in Parallel Branches

| Optimizer | Main Branch | Para. Branch | Train Steps | Main LR | Para. LR | Mean Dice |
|---|---|---|---|---|---|---|
| SGD | $21 \times 21 \times 21$ | $\times$ | 60000 | 0.0005 | $\times$ | 0.849 |
| SGD | $21 \times 21 \times 21$ | $\times$ | 60000 | 0.0004 | $\times$ | 0.852 |
| SGD | $21 \times 21 \times 21$ | $\times$ | 60000 | 0.0003 | $\times$ | 0.856 |
| SGD | $21 \times 21 \times 21$ | $\times$ | 60000 | 0.0002 | $\times$ | 0.859 |
| SGD | $21 \times 21 \times 21$ | $\times$ | 60000 | 0.0001 | $\times$ | 0.854 |
| AdamW | $21 \times 21 \times 21$ | $\times$ | 60000 | 0.0005 | $\times$ | 0.855 |
| AdamW | $21 \times 21 \times 21$ | $\times$ | 60000 | 0.0004 | $\times$ | 0.859 |
| AdamW | $21 \times 21 \times 21$ | $\times$ | 60000 | 0.0003 | $\times$ | 0.861 |
| AdamW | $21 \times 21 \times 21$ | $\times$ | 60000 | 0.0002 | $\times$ | 0.862 |
| AdamW | $21 \times 21 \times 21$ | $\times$ | 60000 | 0.0001 | $\times$ | 0.860 |
| SGD | $21 \times 21 \times 21$ | $3 \times 3 \times 3$ | 60000 | 0.0002 | 0.0006 | 0.872 |
| SGD | $21 \times 21 \times 21$ | $3 \times 3 \times 3$ | 60000 | 0.0002 | 0.0005 | 0.869 |
| SGD | $21 \times 21 \times 21$ | $3 \times 3 \times 3$ | 60000 | 0.0002 | 0.0004 | 0.867 |
| SGD | $21 \times 21 \times 21$ | $3 \times 3 \times 3$ | 60000 | 0.0002 | 0.0003 | 0.870 |
| SGD | $21 \times 21 \times 21$ | $3 \times 3 \times 3$ | 60000 | 0.0002 | 0.0001 | 0.865 |
| AdamW | $21 \times 21 \times 21$ | $3 \times 3 \times 3$ | 60000 | 0.0002 | 0.0006 | 0.887 |
| AdamW | $21 \times 21 \times 21$ | $3 \times 3 \times 3$ | 60000 | 0.0002 | 0.0005 | 0.886 |
| AdamW | $21 \times 21 \times 21$ | $3 \times 3 \times 3$ | 60000 | 0.0002 | 0.0004 | 0.887 |
| AdamW | $21 \times 21 \times 21$ | $3 \times 3 \times 3$ | 60000 | 0.0002 | 0.0003 | 0.889 |
| AdamW | $21 \times 21 \times 21$ | $3 \times 3 \times 3$ | 60000 | 0.0002 | 0.0001 | 0.886 |

To empirically validate the theoretical insight of the spatially varying convergence dynamics in parallel-branched re-parameterization, we initially perform experiments using the CSLA block with Rep3D network architecture, composing of a main large kernel branch ($21 \times 21 \times 21$) and a parallel small kernel branch ($3 \times 3 \times 3$), with separate learning rates applied to each. As shown in Table 3, the single-branch design (no parallel branch) performance improved moderately with lower learning rates with both SGD and AdamW. The Dice score peaks at 0.859 with a learning rate of 0.0002 using SGD, and AdamW achieves its best performance of 0.862 at 0.0002 as well. However, with the addition of a small kernel parallel branch and using a higher learning rate for the small kernel (e.g., $\lambda_S > \lambda_L$), we observed consistent improvements across all configurations. Specifically, the best result with SGD reached 0.872 when using $\lambda_L = 0.0002$ and $\lambda_S = 0.0006$. Similarly, AdamW attained a maximum Dice score of 0.889 with $\lambda_L = 0.0002$ and $\lambda_S = 0.0003$. These results validate our hypothesis that assigning higher learning rates to the small kernel branch accelerates convergence of central kernel regions, while maintaining stability in peripheral regions with a lower learning rate for the large kernel. Moreover, such results further confirm that spatially varying convergence behavior can be approximated through differentiated learning rates, supporting the design principle behind our learnable re-parameterization in Rep3D.

## B ABLATION STUDY ON NETWORK DEPTH FOR LRBM

Table 9: Ablation Study on Network Depth for LRBM with the AMOS testing split

| Number of Layers | Spleen | R. Kid | L. Kid | Gall. | Eso. | Liver | Stom. | Aorta | IVC | Panc. | RAG | LAG | Duo. | Blad. | Pros. | Avg |
|---|---|---|---|---|---|---|---|---|---|---|---|---|---|---|---|---|
| 1 Layer | 0.974 | 0.965 | 0.964 | 0.925 | 0.859 | 0.982 | 0.926 | 0.956 | 0.920 | 0.842 | 0.824 | 0.781 | 0.842 | 0.915 | 0.879 | 0.904 |
| 2 Layers | **0.978** | **0.970** | 0.964 | **0.928** | **0.871** | **0.984** | **0.927** | **0.960** | **0.930** | **0.851** | **0.828** | **0.784** | **0.850** | **0.920** | **0.881** | **0.910** |
| 3 Layers | 0.971 | 0.964 | **0.965** | 0.924 | 0.841 | 0.983 | 0.920 | 0.952 | 0.910 | 0.839 | 0.819 | 0.779 | 0.837 | 0.910 | 0.870 | 0.899 |

## B.1 ADDITIONAL COMPARISONS WITH NNU-NET VARIANTS: RESENC NNU-NET, STU-NET, MEDNEXT

we conducted further comparisons with recent state-of-the-art (SOTA) 3D medical image segmentation architectures, including **ResEnc nnU-Net** Isensee et al. (2021), **STU-Net-H** Huang et al. (2023), and **MedNeXt** Roy et al. (2023). These models have demonstrated strong performance across various benchmarks and provide important context for positioning Rep3D among contemporary architectures.

All baseline methods were trained under their official recommended training schedules (typically 1000 epochs), using the same computational resources and training splits for fair comparison.

| Method | AMOS CT | AMOS MRI | KiTS | Pancreas | Hepatic |
|--------|---------|----------|------|----------|---------|
| nnU-Net (1000 epochs) | 0.887 | 0.847 | 0.706 | 0.703 | 0.660 |
| ResEnc nnU-Net | 0.892 | 0.850 | 0.711 | 0.706 | 0.661 |
| STU-Net-H | 0.900 | 0.848 | 0.707 | 0.712 | 0.648 |
| MedNeXt | 0.897 | 0.856 | 0.720 | 0.713 | 0.663 |
| **Rep3D (Ours)** | **0.910** | **0.864** | **0.736** | **0.723** | **0.674** |

Table 10: Average Dice scores across five segmentation benchmarks under full training schedules. Rep3D consistently outperforms strong baselines across datasets.

These comparisons further validate the strong and consistent performance of **Rep3D** across multiple challenging 3D segmentation benchmarks. Even under extensive training schedules (1000 epochs), Rep3D outperforms the SOTA alternatives, demonstrating its robustness and generalizability.

