# OpenReview forum: "Rep3D: Re-parameterize Large 3D Kernels with Low-Rank Receptive Modeling for Medical Imaging"
_ICLR.cc/2026/Conference — ICLR 2026 Conference Withdrawn Submission_

### Official Review · Reviewer_u2kB · 2025-10-24

**Soundness:** 3
**Presentation:** 3
**Contribution:** 2
**Rating:** 4
**Confidence:** 4

**Summary:**

This paper proposes Rep3D, a framework designed to efficiently train 3D convolutional networks with large kernels through reparameterization. The authors first theoretically derive that the CSLA is equivalent to applying different learning rates to different spatial locations of the convolution kernel. Based on this, the authors propose a module called LRBM. This module uses a lightweight generator network to learn an adaptive spatial modulation mask. This mask is used to adjust the gradient of the large convolution kernel during training, thereby simulating the diffusion behavior of the ERF from local to global. Experimental results show that Rep3D outperforms existing methods on multiple 3D segmentation benchmarks.

**Strengths:**

**Clear Motivation and Solid Theory:** The paper's starting point is clear: to address the question of why naively increasing the size of 3D convolutional kernels is ineffective. The authors go beyond observational analysis and, through mathematical derivation in Section 3.1, provide a strong theoretical basis for the hypothesis that central regions converge faster than peripheral regions. Linking structural reparameterization with spatially variable effective learning rates is a profound insight.

---

**Good Method (LRBM):** The paper's greatest strength lies in transforming this theoretical insight into a learnable module (LRBM). Compared to the fixed priors (e.g., 3D RepUX-Net) or constant scaling (e.g., RepOptimizer) used in previous work, Rep3D's M = P + fθ(P) design (i.e., "fixed prior + learnable dynamic adjustment") is significantly more flexible and powerful. It allows the network to dynamically adjust the gradient contributions of different kernel positions based on the data and task, thereby better simulating the diffusion behavior of the ERF.

---

**Clear Writing and Well-Structured:** The paper flows smoothly, with a clear logical chain from the introduction of the problem, theoretical analysis, method design, to experimental verification. Figures 1 and 2 intuitively demonstrate the differences between Rep3D and traditional methods, which is easy to understand.

**Weaknesses:**

## Overclaimed Novelty

This paper's core motivation (instability of large kernel training), theoretical foundation, and core concepts of its solution (using a distance-based spatial prior to reparameterize/reweight gradients) are nearly identical to those in the **previously published paper "Scaling Up 3D Kernels with Bayesian Frequency Re-parameterization for Medical Image Segmentation" (RepUX-Net).**
The "Bayesian Frequency Reparameterization" (BFR) method proposed in RepUX-Net, which uses a fixed inverse distance function to generate a prior for scaling kernel weights, is methodologically equivalent to the "Rep3D (Fixed Prior)" baseline presented in the experiments of this paper. (I'm not entirely sure of this, but I compared the experimental results and they are completely consistent.) The authors downgraded the core method of their prior work to a baseline for internal ablation experiments without citing or explaining it, which is a highly unscientific practice. Therefore, the **only** innovation of this paper compared to RepUX-Net is the addition of a lightweight learnable generator fθ on top of the fixed spatial prior P, which makes the final modulation mask M = P + fθ(P) adaptive. This downgrades the contribution of this paper from "proposing a completely new framework" to "incremental improvement to an existing framework."

---

## Insufficient Experimental Comparison

Due to the first point mentioned above, this paper's state-of-the-art claim is based on an incomplete baseline comparison. The reader cannot judge the substantial improvement achieved by the proposed learnable module (LRBM) over RepUX-Net's fixed prior (BFR).

Based on the data in Tables 1 and 2, the performance improvement of "Rep3D (LRBM)" over "Rep3D (Fixed Prior)" is present, but the magnitude is limited. Considering that RepUX-Net itself is already state-of-the-art, it is questionable whether this incremental improvement constitutes a significant independent contribution. Furthermore, other models have been proposed since 2023, and the authors do not compare them beyond the baseline model.

---

## Trade-off between High Computational Cost and Effectiveness

As you pointed out, Rep3D's FLOPs (757 G) are high, **exceeding** Transformer models like UNETR (82 G) and SwinUNETR (328 G). While large kernel convolution avoids the quadratic complexity of Attention, a 21x21x21 3D convolution kernel itself carries a significant computational and memory burden. While discussing its advantages, the paper fails to adequately address this significant practical hurdle, undermining the practical value of its approach.

---

## Inappropriate Terminology

**Naming this method "low-rank" is inappropriate.** The LRBM module uses lightweight depthwise separable convolutions and does not apply any mathematically defined low-rank decomposition (such as SVD or Tucker decomposition) to constrain the weights. This is misleading, and we recommend using more accurate terms such as "lightweight generator" or "parameter-efficient."

---

## Others

The paper mentions some limitations in the Discussion section, but does not delve deeply into the core issue of balancing the significant computational cost with performance gains.

**Questions:**

See above critical flaw. I may suggest restructure the narrative logic of the paper. Clearly position the contribution of this paper as "building on RepUX-Net by introducing a learnable modulation module to further improve performance", rather than a completely new framework. Also include other significant baselines. Include detailed discussion of the computational cost. Change the name of "low-rank".

---

### Official Review · Reviewer_ZQhW · 2025-10-31

**Soundness:** 2
**Presentation:** 2
**Contribution:** 1
**Rating:** 2
**Confidence:** 4

**Summary:**

The authors introduce a method Rep3D which is designed to introduce stability during training for large 3D kernels for medical image segmentation. The authors demonstrate that in parallel branches of large and small kernels, the small kernels converging rapidly enhances the performance of the large kernel. Consequently the authors propose a receptive bias re-parameterization strategy that learns a low rank mask to reparameterize the kernel weights during training, while not generating it during testing. This results in state of the art performance against numerous baselines on 5 datasets including organ and tumor datasets.

**Strengths:**

1. The work does introduce novelty in the learnable modulation of the kernel reparameterization.
2. The results demonstrate effectiveness of their reparameterization is superior to the “fixed prior” variant of the algorithm as well as numerous baselines on multiple tasks.

**Weaknesses:**

1. The overlaps with [2] have to be called into question in large parts of the methodology of this work. This is one of the major issues with this work in my opinion and needs addressing.
    1) For example, in Section 3.1 VARIABLE LEARNING CONVERGENCE IN PARALLEL BRANCH, everything till equation 8 is seemingly identical to "Section 3.1 Variable Learning Convergence in Multi-Branch Design" in [2] with just a few additional equations in the subsequent lines in this work which do not necessarily seem relevant. Equation 9, 10, 11 and 12 seem superfluous if the subsequent step is to do something like different learning rates for the branches, and conclude that a higher learning rate in the small kernel branch improves performance for the large kernel branch - a fine finding in general but in [2] there is the line “the locality convergence in large kernels enhance with the quick convergence in small kernels” which makes me question the novelty and ethics of this work.
    2) “Section 3.2 LOW-RANK RECEPTIVE BIAS MODELING (LRBM)” is not unique and the initial part of section 3.2 of this paper is contained in the “Section 3.2 Bayesian Frequency Re-parameterization (BFR)” of [2] while references to [2] in this section specifically are missing.
    3)  While Section 3.3 is identical in content, to Section 3.3 of [2], it is the simple description of the modular architecture and can be possibly excused for overlap on its own but raises questions in conjunction with the points above.

2. While the work closely mirrors experimental design in previous work in [1, 2], there have been popular architectures that have come into prominence since then as described in [3] such as MedNeXt [4] or ResEncL-nnUNet [5] which compete better against the advanced architecture design as presented in this work for medical image segmentation.

3. While perhaps not technically incorrect, the writing feels unnecessarily cumbersome in places almost to the point of hurting readability.

References:

[1] Lee, Ho Hin, et al. "3d ux-net: A large kernel volumetric convnet modernizing hierarchical transformer for medical image segmentation." ICLR 2023.

[2] Lee, Ho Hin, et al. "Scaling up 3d kernels with bayesian frequency re-parameterization for medical image segmentation." MICCAI, 2023.

[3] Bassi, Pedro RAS, et al. "Touchstone benchmark: Are we on the right way for evaluating ai algorithms for medical segmentation?." NeurIPS, 2024.

[4] Roy, Saikat, et al. "Mednext: transformer-driven scaling of convnets for medical image segmentation." MICCAI, 2023.

[5] Isensee, Fabian, et al. "nnu-net revisited: A call for rigorous validation in 3d medical image segmentation." MICCAI, 2024.

**Questions:**

1. The authors should address the concerns regarding the method overlaps in [2]. This is an issue that supersedes novelty or experimental concerns.
2. The results will be more convincing if comparisons against newer architectures as mentioned in the weaknesses such as ResEncL-nnUNet or MedNeXt-L are performed.

References:

[2] Lee, Ho Hin, et al. "Scaling up 3d kernels with bayesian frequency re-parameterization for medical image segmentation." MICCAI, 2023.

**Details Of Ethics Concerns:**

The unacknowledged overlaps with [2] in the methods are too extreme to ignore. The author cites [2] twice in areas distant from the method description, yet derives significant portions of the methods from [2]. If at each step, the author acknowledged that they were merely recreating the analysis from [2], I would feel that the novelty is somewhat weak but would definitely be less inclined to flag this. However, I believe that I need comments on the matter from an AC and/or the authors to clear my objections - that the authors have plagiarized large portions of the methods section of this paper from [2].

Reference

[2] Lee, Ho Hin, et al. "Scaling up 3d kernels with bayesian frequency re-parameterization for medical image segmentation." MICCAI, 2023, https://arxiv.org/abs/2303.05785

---

### Official Review · Reviewer_gdyF · 2025-11-01

**Soundness:** 3
**Presentation:** 2
**Contribution:** 3
**Rating:** 6
**Confidence:** 3

**Summary:**

This paper introduces Rep3D, a 3D convolutional neural network framework that addresses the challenge of training large kernel convolutions for medical image segmentation. The key innovation is a learnable spatial bias re-parameterization strategy that models the diffusion-like behavior of effective receptive fields (ERFs). The authors provide theoretical analysis showing that parallel-branch convolution blocks (e.g., CSLA) implicitly induce spatially varying learning rates, with central kernel elements converging faster than peripheral ones. Building on this insight, Rep3D employs a lightweight two-stage modulation network that generates receptive-biased scaling masks to adaptively re-weight kernel updates during training. The method is evaluated on five challenging 3D medical segmentation benchmarks (AMOS, KiTS, MSD Pancreas, MSD Hepatic Vessel) and demonstrates consistent improvements over state-of-the-art transformer and CNN-based methods.

**Strengths:**

- Strong theoretical foundation: The mathematical analysis linking parallel branches to element-wise learning rates provides valuable insights into why structural re-parameterization works
- Comprehensive evaluation: Testing on 5 different datasets with both CT and MRI modalities demonstrates generalizability
- Consistent improvements: Rep3D shows gains across all benchmarks compared to strong baselines including recent transformer methods
- Practical design: The plain encoder architecture with minimal overhead makes the approach accessible
- Good ablation studies: Thorough investigation of design choices including kernel sizes, network depth, and the effect of LRBM on other architectures

**Weaknesses:**

- Limited novelty: The core components (large kernels, re-parameterization, ERF modeling) are well-established; the combination is somewhat incremental
- Computational cost not fully addressed: While parameter counts are provided, actual training time and memory comparisons are limited to a small ablation in the appendix
- Missing comparisons: No comparison with some recent methods like UniverSeg or other foundation models for medical imaging
- Unclear practical impact: The improvements, while consistent, are often modest (1-3% Dice score) - clinical significance is not discussed
- Implementation details: Critical details like how the mask is removed at inference, exact training procedures, and convergence criteria are vague
- Limited analysis of failure cases: No discussion of when/why the method might fail or perform poorly

**Questions:**

1. Inference efficiency: How exactly is the modulation mask removed during inference? Does this require any special handling or re-training?

2. Memory requirements: What are the actual GPU memory requirements for training with 21×21×21 kernels? How does this compare to transformer baselines?

3. Generalization to other domains: Have you tested this approach on natural images or other 3D vision tasks beyond medical imaging?

4. Clinical significance: The improvements are often 1-3% in Dice score. Have you consulted with medical professionals about whether these improvements are clinically meaningful?

5. Theoretical gap: The theory assumes SGD but experiments use AdamW. While you provide some discussion in the appendix, could you elaborate on how adaptive optimizers affect your theoretical insights?

6. Distance decay function: How sensitive is the method to the specific form of the distance decay function (Equation 14)? Have you tried other spatial priors?

7. Comparison with attention: Given that transformers also exhibit center-focused attention patterns, could you provide more direct comparisons or analysis of the differences?

8. Training stability: You mention optimization instability with large kernels. Can you provide training curves comparing Rep3D with baseline methods to demonstrate improved stability?

---

### Official Review · Reviewer_Tknj · 2025-11-05

**Soundness:** 2
**Presentation:** 1
**Contribution:** 2
**Rating:** 4
**Confidence:** 5

**Summary:**

This paper proposes a new method to learn large-scale convolutional kernels for very high resolution tensors, such as medical data. The so-called Rep3D is based on a learnable weighting of the convolution kernel. Qualitative analysis demonstrates a large-scale scale effective receptive field. The approach is validated on numerous medical imaging datasets.

**Strengths:**

The main strength of this method is its simplicity and potential application for a broad spectrum of applications.

The approach proves to be competitive even against top baselines.

The receptive field enhancement hypothesis is validated through a dedicated ablation study.

**Weaknesses:**

The paper might lack original content, which tends to be highlighted by a too long preliminary section on the chain rule. The idea of extending the receptive field of convolutional kernels dates back to deformable convolutions [1]. It has been an active line of work in computer vision for the past few years. Several recent works have been proposed to enhance and stabilize the receptive field of large kernels. Recently, Pelk [2] introduced parameter-efficient large kernels based on a peripheral vision mechanism, thus sharing the same philosophy as Rep3D. The proposed Rep3D method lacks positioning and comparison with these new approaches.



[1] Dai, Jifeng, et al. "Deformable convolutional networks." Proceedings of the IEEE international conference on computer vision. 2017.
[2] Chen, Honghao, et al. "Pelk: Parameter-efficient large kernel convnets with peripheral convolution." Proceedings of the IEEE/CVF conference on computer vision and pattern recognition. 2024.

**Questions:**

l. 239-249: the conclusion that the weights for the central region of the kernel converge the fastest is not in agreement with the learning rates associated with the best performances where $\lambda_S < \lambda_L$.

What are the $\beta$ values after training, and does it confirm the intuition on the kernels' convergence?

What is the magnitude of $f_\theta$ compared to P, and does this function really behave as a residual? Nothing prevents the model to learn a mapping in the form of $f_\theta(P) = -P +  f'_\theta(P)$.

It is unclear what is the parameter overhead due to the $f_\theta$ function. Are the $\theta$ parameters shared across the architecture?

The qualitative evaluation of the effective receptive field size is difficult to assess and would benefit from a quantitative evaluation.

---

### Note · Authors · 2025-11-21

I have read and agree with the venue's withdrawal policy on behalf of myself and my co-authors.